# Preparation and Lithium-Ion Separation Property of ZIF-8 Membrane with Excellent Flexibility

**DOI:** 10.3390/membranes13050500

**Published:** 2023-05-09

**Authors:** Jun Zhao, Rongyu Fan, Shengchang Xiang, Jiapeng Hu, Ximing Zheng

**Affiliations:** 1School of Chemistry and Materials, Fujian Normal University, Fuzhou 350001, China; 2Fujian Provincial Key Laboratory of Eco-Industrial Green Technology, Key Laboratory of Green Chemical Technology of Fujian Province University, Wuyi University, Wuyishan 354300, China

**Keywords:** lithium-ion selectivity, ion permeation flux, metal-organic frameworks, polypropylene membrane, ZIF-8 membrane, flexibility

## Abstract

Metal-organic framework (MOF) membranes exhibit immense potential for separation applications due to their regular pore channels and scalable pore sizes. However, structuring a flexible and high-quality MOF membrane remains a challenge due to its brittleness, which severely restricts its practical application. This paper presents a simple and effective method in which continuous, uniform, defect-free ZIF-8 film layers of tunable thickness are constructed on the surface of inert microporous polypropylene membranes (MPPM). To provide heterogeneous nucleation sites for ZIF-8 growth, an extensive amount of hydroxyl and amine groups were introduced on the MPPM surface using the dopamine-assisted co-deposition technique. Subsequently, ZIF-8 crystals were grown in-situ on the MPPM surface using the solvothermal method. The resultant ZIF-8/MPPM exhibited a lithium-ion permeation flux of 0.151 mol m^−2^ h^−1^ and a high selectivity of Li^+^/Na^+^ = 1.93, Li^+^/Mg^2+^ = 11.50. Notably, ZIF-8/MPPM has good flexibility, and the lithium-ion permeation flux and selectivity remain unchanged at a bending curvature of 348 m^−1^. These excellent mechanical characteristics are crucial for the practical applications of MOF membranes.

## 1. Introduction

Lithium is in high demand as a new energy and strategic resource. The total lithium resource from salt lake brines and seawater is substantially greater than that in lithium ore, which is considered an important means to obtaining lithium resources both at present and in the future [1]. The current lithium separation and extraction techniques include adsorption [2], extraction [3], membrane separation [4], and electrochemical methods [5]. Among these, membrane separation, which is a method of lithium extraction by physical means, has been extensively studied for its advantages, such as its simple operation process, lack of phase change in separation, and lack of chemical additives [6,7,8,9]. In salt lake brines and seawater, lithium ions are usually mixed with alkali metal ions and alkaline earth metal ions of comparable ionic sizes [10,11], posing a major challenge to conventional separation membranes with a wide channel size distribution.

Metal-organic frameworks (MOFs) are structurally ordered inorganic porous materials synthesized from secondary structural units composed of metal ions or clusters and organic ligands. As a result of their regular pore structure, as well as their precisely adjustable pore size, MOFs are ideal film-forming materials for designing high-quality lithium ion separation membranes [12]. Nevertheless, the formation of a membrane using only MOF materials is challenging. At present, the reported MOF membranes mainly include substrate-supported heteroepitaxial membranes and mixed matrix membranes formed by combining MOF and polymers [13,14,15]. In mixed matrix membranes, the gaps between the aggregated MOF particles and/or MOF-polymer interface defects provide an ion transport path that bypasses the selectivity framework, which results in low ion selectivity [12,16,17]. A substrate-supported homogeneous MOF membrane is a more desirable material for ion separation. Xu et al. [16] achieved the construction of a dense UiO-66-NH_2_ blade-like layer on the surface of an anodic alumina oxide (AAO) substrate using a counter-diffusion growth method. The nano-pores, as well as the Å size pores, in the membrane layer ensure fast ion transport and selective sieving, which exhibit excellent selectivity of Na^+^/Mg^2+^ > 200, Li^+^/Mg^2+^ > 60. Guo et al. [17] introduced sodium polystyrene sulfonate into HKUST-1 to construct a PSS@HKUST-1 layer on the surface of an AAO substrate, and the introduction of sodium polystyrene sulfonate not only significantly improved the water stability of the HKUST-1, but also formed a three-dimensional network of sulfonic acid for ion transport. Due to the affinity difference between metal-ion and sulfonic acid groups and the synergistic effect of pore size sieving, PSS@HKUST-1 exhibited a lithium-ion permeation flux of 6.75 mol m^−2^ h^−1^ and an ultra-high selectivity of Li^+^/Na^+^ = 35, Li^+^/K^+^ = 67, and Li^+^/Mg^2+^ = 1815. The ZIF-8 is a metal-organic backbone with a 2-methylimidazole ligand coordinating the zinc ion center and has a 0.34 nm pore window and a pore cavity 1.16 nm in diameter [18]. The special pore structure and pore size of ZIF-8 are expected to be instrumental in the construction of highly selective lithium-ion separation membranes [19,20]. Wang et al. [20] constructed dense GO@ZIF-8 layers on the inorganic substrate AAO surface for the separation of lithium ions out of other alkali metal ions through spin-coating and counter-diffusion growth. It was observed that the selectivity of Li^+^/Rb^+^ reached 4.6, which is significantly higher than the Li^+^/Rb^+^ selectivity measured in conventional separation membranes. The industrial application of ZIF-8 membranes is limited by the drawbacks of inorganic substrates, such as their difficult processing and inability to bend.

Polymer substrates offer flexible support for ZIF-8 film layer, which makes them better suited for installation applications. However, the lack of heterogeneous sites makes it difficult to construct a high-quality ZIF-8 membrane onto the polymer substrate. To the best of our knowledge, only a few works about constructing a ZIF-8 film layer on polymer substrates have been reported to date [21,22,23]. For CO_2_/N_2_ separation, Meng et al. [21] successfully prepared a ZIF-8 film layer on a PSF substrate by embedding ZIF-8 crystals in the substrate as anchors linking the ZIF-8 film layer with the substrate, and the performance was unaltered after bending the membrane with a curvature of 36.3 m^−1^. For H_2_/CO_2_ separation, Liu et al. [22] also synthesized a ZIF-8 membrane on porous polyether sulfone fiber by embedding a seed crystal, and the membrane presented a good separation performance even after being bent at about 46 m^−1^. For the separation of propylene propane, Wang et al. [23] prepared a conductive layer by coating platinum onto the polymer substrate, and then constructed a ZIF-8 film on polypropylene membranes through electrochemical deposition, and the performance remained unaffected after membrane bending at 92 m^−1^. Hitherto, polymer-supported ZIF-8 membranes have mainly been used for gas separation, while they have seldom been reported to separate ion.

A polypropylene membrane is able to offer ideal flexible support for ZIF-8 in view of their Young’s modulus similarity. Herein, a microporous polypropylene membrane (MPPM) was applied to construct a ZIF-8 membrane as the substrate. The surface of the MPPM was enriched with hydroxyl and amine groups through dopamine-assisted polyvinylimineco deposition. Then, Zn(II) was fixed onto the membrane surface through the complexation between the Zn(II) and amine group, as well as the hydroxyl group, which provided chemically bonded heterogeneous nucleation sites for the growth of ZIF-8 on the membrane. Finally, a continuous, uniform, and defect-free ZIF-8 film layer was constructed on the MPPM surface using the solvothermal method. The fabricated ZIF-8 membranes exhibited an efficient Li^+^ separation performance and excellent flexibility. The performance remained unaffected even when subjected to large bending caused by external stress. The method for constructing the ZIF-8 membrane is not only simple, but also highly reproducible and scalable. It is able to provide reference and inspiration for the development of high-performance ion separation films. 

## 2. Materials and Methods

### 2.1. Materials

The microporous polypropylene membrane (MPPM; average pore size = 0.2 µm, porosity = 75%) was purchased from Membrana Corp. Dopamine hydrochloride (DA, ≥98%), polyethyleneimine (PEI, Mw = 600), 2-methylimidazole (C_4_H_6_N_2_), sodium formate (HCOONa), lithium chloride (LiCl), and sodium chloride (NaCl) were purchased from Maclean’s Reagent. Magnesium chloride hexahydrate (MgCl_2_·6H_2_O) and Zinc nitrate hexahydrate (Zn(NO_3_)_2_·6H_2_O) were purchased from the Chinese medicine reagent (Shanghai, China). All chemicals were of analytical grade and were not further purified.

### 2.2. Surface Modification of MPPM

To form a 2 mg/mL solution for each, DA and PEI were dissolved in Tris-HCl buffer solution (pH = 8.5). The ethanol-impregnated MPPM was submerged into the aforementioned solution and shaken for 12 h at room temperature. Subsequently, it was removed and rinsed with deionized water for 6 h through shaking, dried under vacuum at 40 °C, and placed in a dryer for standby.

### 2.3. Construction of ZIF-8 Membrane

MPPM@PDA-PEI was submerged into a methanol solution of Zn(II) (0.1 M) for 1 h to complex and adsorb Zn(II) onto the surface of MPPM. To make the ZIF-8 precursor solution, 0.8925 g zinc nitrate hexahydrate together with 0.255 g sodium formate, and 0.4926 g 2-methylimidazole were ultrasonically dissolved in 15 mL methanol, respectively, and then the two 15 mL prepared solutions were combined. The obtained precursor solution of ZIF-8 was poured into the reaction kettle. Furthermore, the Zn(II)-complexed MPPM was placed face up at the bottom of the kettle. After 16 h of reaction at 60 °C, the membrane was wiped off and rinsed using methanol, dehydrated at room temperature, and placed in a dryer for standby.

### 2.4. Characterization

To analyze the chemical structure of the membranes, attenuated total reflection Fourier transform infrared spectroscopy (ATR-FTIR, Nicolet iS5, Thermo Fisher Scientific, Waltham, MA, USA) was employed in the wavelength range of 4000–400 cm^−1^. The surface and cross-sectional morphologies of the membranes were characterized using field emission scanning electron microscopy (SEM, TESCAN VEGA3 SBH, Brno, Czech Republic). The phase ZIF-8 membrane structure was determined using X-ray diffraction (XRD, D8 ADVANCE, Karlsruhe, Germany) with Cu Kα radiation from 2° to 60° at a 10° min^−1^ scan rate. Using an ESCA system with monochromatic Mg Kα as the radiation source, the elemental composition and electron valence band positions of the samples were determined using X-ray photoelectron spectroscopy (XPS, K-Alpha+ model origin, Thermo Fisher Scientific, Waltham, MA, USA).

### 2.5. Ion Separation Performance Test

Ion permeation testing was performed using a homemade H-type electrolytic cell. The ZIF-8/MPPM membrane was fastened between the two half electrolytic cells to prevent leakage using two silicone gaskets with center openings 15 mm in diameter. The unitary ion permeation test was performed using a single solution of 0.1 M LiCl, 0.1 M NaCl, or 0.1 M MgCl_2_ on the feed side. The binary ion permeation test was performed using a mixture of 0.1 M LiCl + 0.1 M NaCl or 0.1 M LiCl + 0.1 M MgCl_2_ on the feed side. The other side for either the unitary ion permeation test or the binary ion permeation test is the receiving chamber full of deionized water. Meanwhile, magnetic stirring was conducted on both sides of the H-type electrolytic cell during the test to relieve the concentration polarization effect on the test results. In the unit ion permeation experiment, samples were taken every 10 min, and a conductivity meter was applied to test the metal ion concentration in the receiving chamber. Each sample was tested thrice to calculate the mean value. The method of determining the ion concentrations from ionic conductivity is based on previous studies [24,25,26]. In the binary ion permeation experiment, the metal ion concentration was determined using an atomic absorption spectrophotometer (AAS, Shimadzu AA-6880, Kyoto, Japan). The equation for the ion permeation flux is as follows:(1)$$J_{i}=\frac{\left(C_{t}-C_{0}\right)\times V}{A_{m}\times t}$$
where *J_i_* denotes the ion permeation flux (mol m^−2^ h^−1^); *C_t_* denotes the current ions concentration in the receiving chamber (mol L^−1^); *C*_0_ denotes the concentration of initial ions in the receiving chamber (mol L^−1^); *V* denotes the receiving chamber solution volume (L); *A_m_* denotes the effective area through the membrane (m^−2^); *t* denotes the test time (h).

The ion selectivity is computed using an equation as follows:(2)$$S=\frac{J_{i}}{J_{j}}$$

## 3. Results and Discussion

### 3.1. Construction of ZIF-8/MPPM

Figure 1 exhibits the construction process of the ZIF-8 on the inert MPPM surface. The MPPM was immersed in a solution of dopamine and low-molecular-weight PEI. Due to oxygen being dissolved in the solution, the dopamine was self-polymerized into polydopamine (PDA). Meanwhile, the MPPM became rich in amine groups and hydroxyl groups as a consequence of the PDA and PEI forming a copolymer via Michael addition or Schiff base reaction and stably adhering to the MPPM surface [27]. The membrane was then immersed in the Zn(II) solution, where the Zn(II) was uniformly and stably adsorbed onto the membrane surface by complexing with the amine groups and hydroxyl groups to form heterogeneous nucleation sites. Finally, the film was submerged into the ZIF-8 precursor solution with Zn(II), as well as 2-methylimidazole, and heated. The complexed Zn(II) on the surface of the MPPM was coordinated with the deprotonated 2-methylimidazole in the solution to form crystal nuclei, which continued to grow on the membrane surface. The crystals were embedded and symbiotic with each other. Eventually, a continuous, uniform, and stable ZIF-8 film was successfully constructed onto the MPPM surface.

SEM characterization of the membranes was performed in the construction analysis of the ZIF-8 membranes. As shown by the comparison between Figure 2a,b, the surface morphology of the films was maintained, undergoing modification through PDA-PEI co-deposition. Figure 2c depicts the surface morphology of the ZIF-8 membrane prepared through in-situ synthesis. Here, the ZIF-8 crystals grow to be embedded with one another to produce a tight, continuous, and defect-free film layer. As displayed in the magnified image in Figure 2d, the ZIF-8 film is tightly embedded in the MPPM, as opposed to being simply sedimented onto the MPPM surface. Moreover, the ZIF-8 crystal does not fall off the MPPM when rubbed by hand.

As seen from the results in Figure 3a–l, the ZIF-8 film layer is significantly impacted by the concentration of sodium formate applied for deprotonation. The ZIF-8 crystals loaded onto the surface of the MPPM without sodium formate were small (Crystal Size, 2.5 µm) and sparsely distributed, and they were easily detached from the MPPM. At a concentration of 0.05 M of sodium formate, the size of the ZIF-8 crystals loaded onto the MPPM surface enlarged (Crystal Size, 8.3 µm), and the distribution was tighter. However, the ZIF-8 film layer presented a large number of defects. When the sodium formate concentration reached 0.1 M, the ZIF-8 crystals were observed to combine with each other on the surface of the MPPM, forming a continuous and dense ZIF-8 film layer., with a thickness of 46 µm (Crystal Size, 38.4 µm). The ZIF-8 film layer remained continuous and dense as the concentration of sodium formate rose to 0.125 M and 0.15 M, and the thickness correspondingly increased to 56 µm and 63 µm. However, the ZIF-8 crystals agglomerate, and a lot of defects arise in the ZIF-8 film layer when the concentration of sodium formate reaches 0.2 M. Moreover, the concentration of the ZIF-8 precursor (constant concentration ratio) affects the growth of the ZIF-8 membrane, which is depicted in Figure 4a–e. At low concentrations of 0.04 M:0.08 M:0.04 M of Zn^2+^:2-methylimidazole:sodium formate, the size of the ZIF-8 crystals was small (Crystal Size, 7.6 µm), irregular, and uneven distribution. The ZIF-8 crystal grows as the concentration increases (the ZIF-8 crystals size increases to 28.8 µm at concentrations of 0.08 M:0.16 M:0.08 M of Zn^2+^:2-methylimidazole:sodium formate), and its distribution becomes more compact and continuous. Nevertheless, the concentration must not be too high, as this would cause the crystals to agglomerate, resulting in a significant number of defects in the ZIF-8 film layer.

### 3.2. Characterization of ZIF-8/MPPM

The membrane was characterized using ATR-FTIR. As depicted in Figure 5a, the absorption band at 1592  cm^−1^ results from the stretching vibration of the C=N bond, the peak at 1145 cm^−1^ is derived from the C-N stretching vibration on the imidazole ring, the absorption band at 755 cm^−1^ belongs to the C-H bending vibration on the imidazole ring, and the peak at 681 cm^−1^ corresponds to the stretching vibration of the ZnO bond in the octahedral coordination [23,28]. Figure 5b displays the XRD characterization of the membranes. The XRD spectrograms of three ZIF-8/MPPM samples revealed the characteristic peaks at 7.47°, 10.53°, 12.83°, 14.84°, 16.69°, and 18.19°, demonstrating the synthesis of the ZIF-8 film layer [29].

Figure 6 shows the XPS characterization of the membranes. The C 1s spectrum exhibited a peak at 286.4 eV, indicating that the carbon linked to N in the methyl imidazole group [30]. The N 1s XPS spectrum revealed two peaks with centers at 401 eV and 399 eV, corresponding to C=N and C-N in 2-methylimidazole, respectively. The Zn 2p_3/2_ and 2p_1/2_ peaks at 1022 eV and 1045 eV indicate that most of the Zn are in the tetrahedral coordination, which agrees with previous reports [23].

### 3.3. ZIF-8/MPPM Ion Separation Performance

The cation permeation properties of the ZIF-8/MPPM were assessed using a homemade H-type electrolytic cell (see Figure 7a for the test setup), with one side filled with metal ion solution and the other side full of deionized water. Figure 7b,c display the results. The order of the ion permeation flux of the unit ions for MPPM@PDA-PEI was Na^+^ > Li^+^ > Mg^2+^, and the ion permeation fluxes of Li^+^, Na^+^, and Mg^2+^ were 7.91 mol m^−2^ h^−1^, 10.81 mol m^−2^ h^−1^, 6.39 mol m^−2^ h^−1^. As per the findings, the ion permeation fluxes are consistent with the sequence of the hydrated ion diameter sizes (Dh) of Na^+^ (Dh = 7.16 Å), Li^+^ (Dh = 7.64 Å), and Mg^2+^ (Dh = 8.24 Å) [31]. Their lithium-ion selectivities were Li^+^/Na^+^ = 0.73 and Li^+^/Mg^2+^ = 1.23, respectively. This is because the diameter of the ion channel on MPPM@PDA-PEI is considerably larger than the size of the hydrated cations. Moreover, when driven by an electric field, the hydrated cations in the solution can easily pass through the MPPM@PDA-PEI channel. ZIF-8/MPPM exhibits excellent Li^+^ selectivity compared to MPPM@PDA-PEI. The metal ion permeation flux of ZIF-8/MPPM exhibited the following order in both the unitary and binary ion separation tests: Li^+^ > Na^+^ > Mg^2+^. For the unitary ion solution, the ion permeation fluxes of Li^+^, Na^+^, and Mg^2+^ were 0.1171 mol m^−2^ h^−1^, 0.0640 mol m^−2^ h^−1^, and 0.0130 mol m^−2^ h^−1^, respectively. In addition, their Li^+^/Na^+^ and Li^+^/Mg^2+^ selectivities were 1.83 and 8.99, respectively. In the binary separation system, competition and interference between ions affect the ion migration via ZIF-8/MPPM, resulting in a reduced selectivity. The selectivity of Li^+^/Na^+^ and Li^+^/Mg^2+^ remained at 1.34 and 5.63, respectively. According to the aforementioned experimental results, ZIF-8/MPPM exhibits excellent selectivity for the ion permeation flux of Li^+^.

In the process of permeation, ions first have to overcome the pore size barrier to enter the membrane pore, and then overcome the internal resistance of the membrane pores to penetrate through the membrane [26,32]; thus, the membrane morphology and test conditions may affect the ion permeation flux. Figure 8a demonstrates that when the thickness of the ZIF-8 film layer was varied from 46 µm to 56 µm, the Li^+^/Mg^2+^ and Li^+^/Na^+^ selectivity significantly increased, but the ion permeation fluxes did not change much. When the thickness of the ZIF-8/MPPM film layer increases from 56 µm to 63 µm, the selectivity of Li^+^/Mg^2+^ stops increasing, and the selectivity of Li^+^/Na^+^ continues to increase slowly, but the ion permeation fluxes decrease significantly. As shown in Figure 8b, with the increase in the pH value from 4.0 to 5.5, the ion permeation fluxes and selectivity of ZIF-8/MPPM increase, the ion permeation fluxes of Li^+^, Na^+^ and Mg^2+^ increase from 0.041 mol m^−2^ h^−1^, 0.030 mol m^−2^ h^−1^ and 0.010 mol m^−2^ h^−1^ to 0.117 mol m^−2^ h^−1^, 0.064 mol m^−2^ h^−1^ and 0.013 mol m^−2^ h^−1^, and the selectivity of Li^+^/Na^+^ and Li^+^/Mg^2+^ increase from 1.38 and 3.99 to 1.83 and 8.99. Although the ion permeation fluxes increase slightly as the pH value continues to increase to 6.5, the selectivity of Li^+^/Na^+^ and Li^+^/Mg^2+^ decline slightly. Figure 8c demonstrates that the ion permeation fluxes of Li^+^, Na^+^ and Mg^2+^ increase when the temperature increases from 288 K to 303 K. The selectivity of Li^+^/Na^+^ is almost unchanged, but the selectivity of Li^+^/Mg^2+^ decreases significantly due to the stronger dependence of Mg^2+^ on temperature [33]. The ion permeation fluxes of Li^+^, Na^+^ and Mg^2+^ versus the temperature on ZIF-8/MPPM was found to agree with the Arrhenius relationship (Figure 8d), and the energy barriers (Ea, activation energy) of Li^+^, Na^+^ and Mg^2+^ were 4.46 kcal mol^−1^, 5.64 kcal mol^−1^ and 13.87 kcal mol^−1^, respectively. Furthermore, Li^+^ has a relatively low transmembrane energy barrier and can easily enter the membrane channel. Considering the ion permeation flux and selectivity, the ZIF-8/MPPM prepared using this method showed a satisfactory ion separation performance (Table 1).

### 3.4. Bending Performance of ZIF-8/MPPM

The mechanical properties of membranes are crucial in real-world applications. The flexibility tests of ZIF-8/MPPM were analyzed at three different bending levels, and the results are displayed in Figure 9a–g. At a bending curvature of 444 m^−1^, ZIF-8/MPPM was bent without obvious cracks, breakage, peeling, or flaking of the membrane layer. The ion permeation fluxes and selectivity of the Li^+^/Mg^2+^ of ZIF-8/MPPM at bending curvatures up to 348 m^−1^ in the ion separation performance test were unaltered compared to those before bending. The ion permeation fluxes of both Li^+^ and Mg^2+^ increased at the bending curvature of 444 m^−1^ compared to those before bending, suggesting that invisible defects should have appeared within the membrane layer. Thus, the selectivity of ZIF-8/MPPM for Li^+^/Mg^2+^ was reduced, but still reached 5.31. The findings reveal that the MPPM-supported ZIF-8 membrane exhibits strong practical application value due to its good flexibility and property retention ability under external stress. Compared with the literature data of the existing ZIF-8 homogeneous film, the ZIF-8/MPPM prepared using this method has the best flexibility (Table 2).

## 4. Conclusions

In this paper, continuous, uniform, and defect-free ZIF-8 film layers were successfully constructed on an MPPM surface using the dopamine-assisted co-deposition technique and the solvothermal method. This method enables the preferential regulation of the ZIF-8 film layer thickness by adjusting the construction parameters. The ion selectivity order of ZIF-8/MPPM is as follows: Li^+^ > Na^+^ > Mg^2+^. The prepared ZIF-8/MPPM possesses good flexibility. The Li^+^/Mg^2+^ separation performance was unaffected when the film was bent for 348 m^−1^. This excellent mechanical property substantially facilitates its practical application. The novel ZIF-8/MPPM membrane synthesis strategy provided in this study can provide a new incentive to develop advanced ZIF membranes. The method for constructing a ZIF-8 membrane is not only simple, but also highly reproducible and scalable. It is able to provide reference and inspiration for the development of high-performance ion separation membranes. 

## Figures and Tables

**Figure 1 membranes-13-00500-f001:**
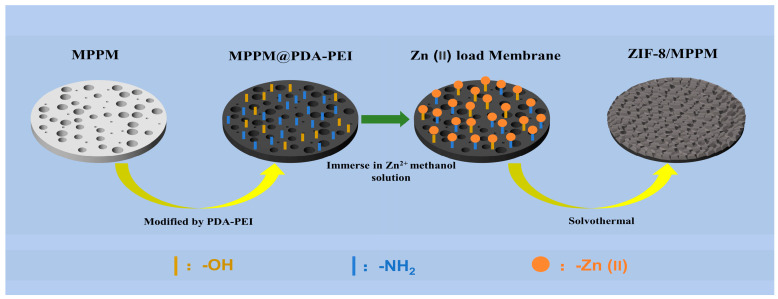
Schematic diagram of ZIF-8/MPPM construction.

**Figure 2 membranes-13-00500-f002:**
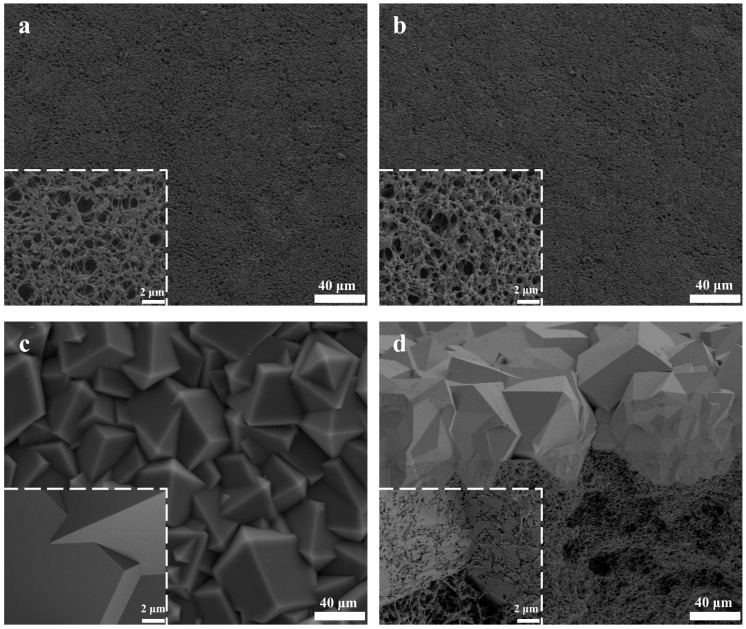
(**a**) SEM images of MPPM; (**b**) SEM images of the MPPM@PDA-PEI; (**c**) SEM images of ZIF-8/MPPM (56 µm); (**d**) Cross section image of the ZIF-8/MPPM (56 µm). The inset is the high magnification image. The 56 µm in ZIF-8/MPPM (56 µm) means the thickness of the ZIF-8 film layer.

**Figure 3 membranes-13-00500-f003:**
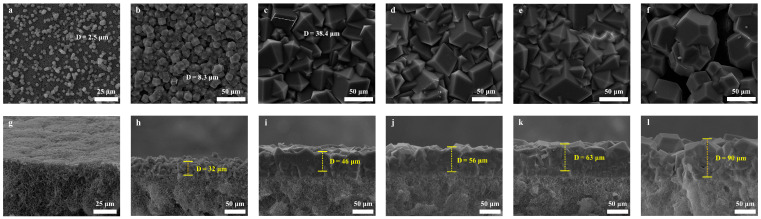
SEM images and cross-sectional SEM images of ZIF-8/MPPM synthesized with different concentrations of sodium formate (the concentrations of Zn^2+^ and 2-methylimidazole is 0.1 M and 0.2 M, respectively,): (**a**,**g**) 0 M; (**b**,**h**) 0.05 M; (**c**,**i**) 0.1 M; (**d**,**j**) 0.125 M; (**e**,**k**) 0.15 M; (**f**,**l**) 0.2 M.

**Figure 4 membranes-13-00500-f004:**
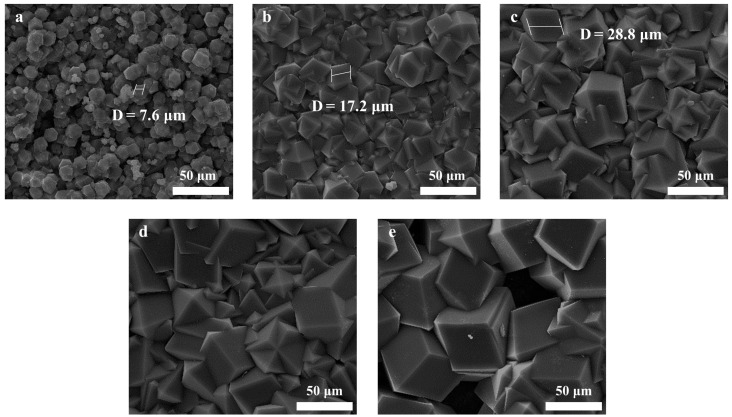
SEM images of ZIF-8/MPPM synthesized with different concentrations of precursor (the concentrations of Zn^2+^:2-methylimidazole:sodium formate is, respectively: (**a**) 0.04 M:0.08 M:0.04 M; (**b**) 0.06 M:0.12 M:0.06 M; (**c**) 0.08 M:0.16 M:0.08 M; (**d**) 0.1 M:0.2 M:0.1 M; (**e**) 0.15 M:0.3 M:0.15 M).

**Figure 5 membranes-13-00500-f005:**
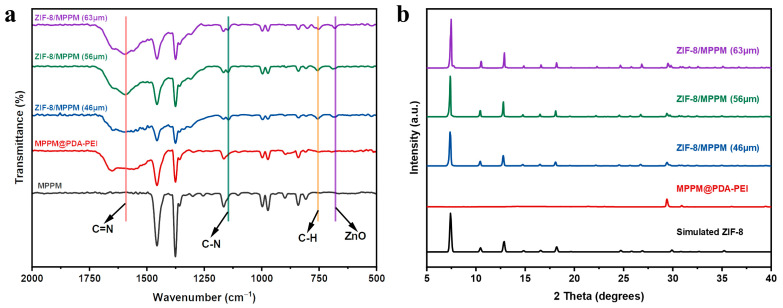
(**a**) ATR-FTIR and (**b**) XRD patterns of the MPPM, MPPM@PDA-PEI and ZIF-8/MPPM with different thickness.

**Figure 6 membranes-13-00500-f006:**
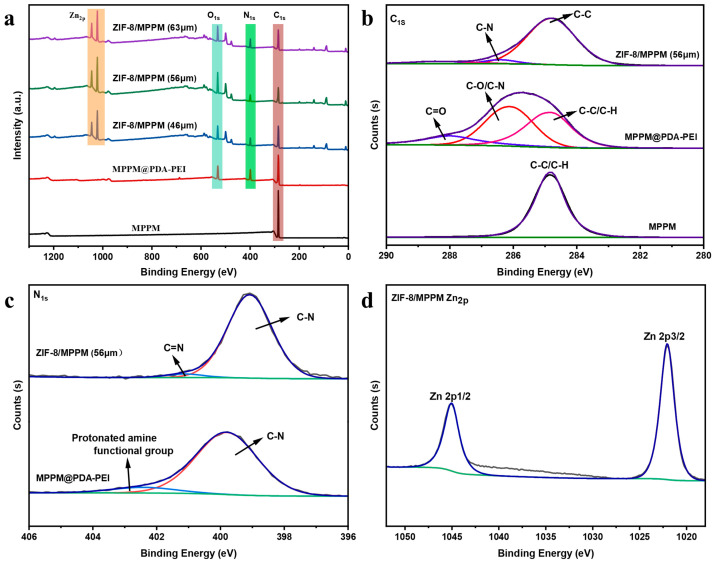
XPS spectra of the MPPM, MPPM@PDA-PEI and ZIF-8/MPPM with different thickness. (**a**) The survey spectrum; (**b**) XPS spectra of C 1s; (**c**) XPS spectra of N 1s; (**d**) XPS spectra of Zn 2p.

**Figure 7 membranes-13-00500-f007:**
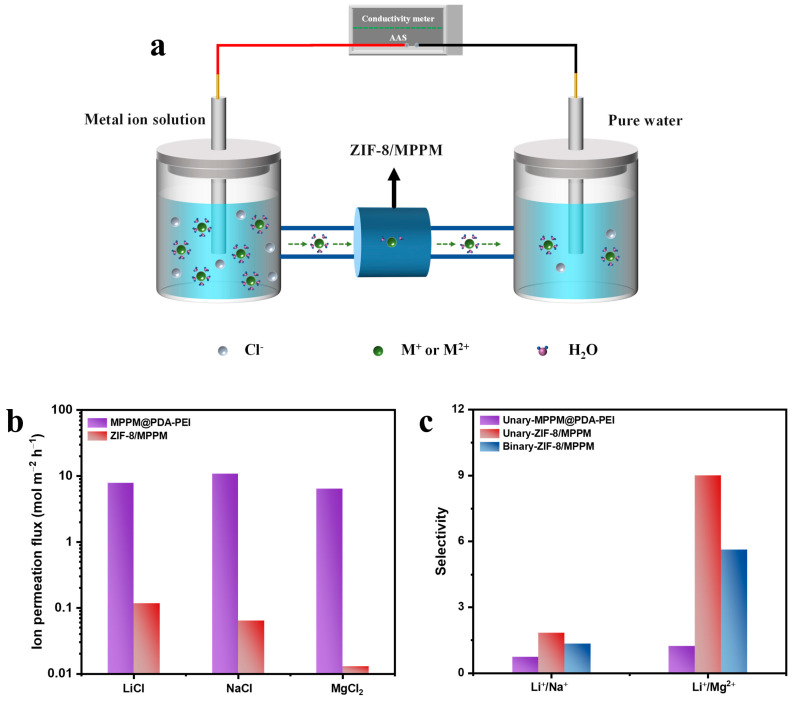
(**a**) Schematic diagram of the self-made H-type electrolytic cell. (**b**) Ion permeation flux of MPPM@PDA-PEI, ZIF-8/MPPM (56 μm) in unary ion permeation test (pH = 5.5, 20 °C). (**c**) Unary ion selectivity (pH = 5.5, 20 °C) of MPPM@PDA-PEI, unary and binary selectivity (pH = 5.5, 20 °C) of ZIF-8/MPPM (56 μm).

**Figure 8 membranes-13-00500-f008:**
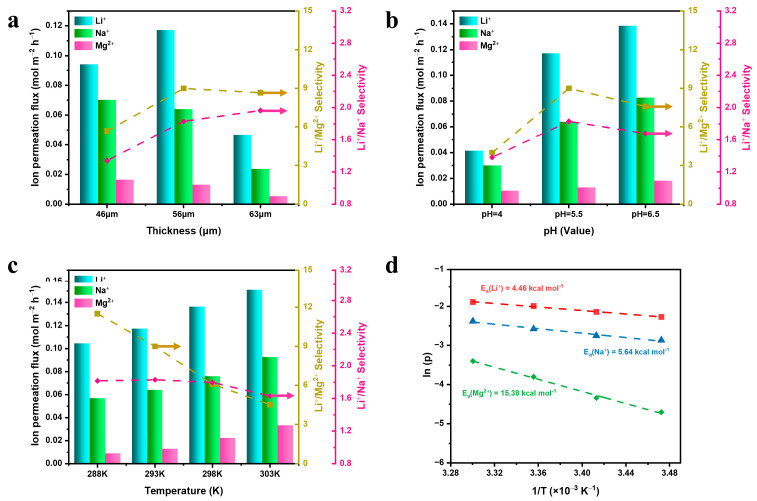
Ion permeation flux and selectivity of ZIF-8/MPPM (**a**) at different thicknesses (pH = 5.5, 20 °C). (**b**) at different pH (56 µm, 20 °C). (**c**) at different temperatures (56 µm, pH = 5.5). (**d**) Arrhenius plots for the ZIF-8/MPPM (56 µm, pH = 5.5).

**Figure 9 membranes-13-00500-f009:**
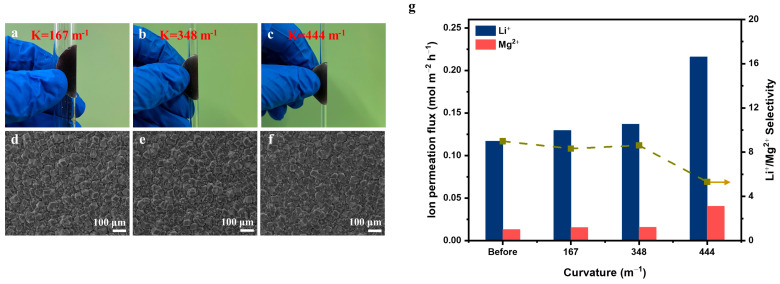
(**a**–**c**) Photographs of the ZIF-8/MPPM with different curvature. The membranes are rolled around cylindrical bars of different diameters. Top-view (**d**–**f**) after bending with different curvature, respectively. K is the curvature of bending. (**g**) Ion permeation flux and selectivity of Li^+^, Mg^2+^ after ZIF-8/MPPM bending.

**Table 1 membranes-13-00500-t001:** The comparison of lithium-ion separation membranes performance.

Membrane	Ion Permeation Flux Li^+^ (mol m^−2^ h^−1^)	Selectivity		Reference
		Li^+^/Na^+^	Li^+^/Mg^2+^	
ZIF-8-0.6@PVC	-	1.34	2.02	[14]
HSO_3_-UiO-66@PVC	8.84 × 10^−3^	-	4.78	[14]
PSS@HKUST-1	6.75	35	1815	[17]
ZIF-8/GO/AAO	-	1.37	-	[20]
MXene	2 × 10^−3^	-	3.07	[25]
UiO-67	27.01	2.02	159.4	[33]
TA-Fe^III^/ZIF-8	-	~0.97	3.87	[34]
ZIF-8-Epoxy	-	-	6.1	[35]
UiO-66-SO_3_H	7.56 × 10^−2^	-	19	[36]
TpBDMe_2_	5.53 × 10^−2^	0.32	217	[37]
UiO-66-(COOH)/UiO-66-NH_2_	-	-	90.8	[38]
MXene	0.01	-	2.0	[39]
PCGO	5 × 10^−3^	1.19	500	[40]
Al13-Ti3C2Tx	0.01	-	2.5	[41]
Sulfonated-GO	1.3	-	8.46	[42]
rGO@SAPS-1	0.06	-	3.8	[43]
FRGO	0.015	-	12	[44]
Prussian white	0.18	-	8.57	[45]
IGM	1.02	2.52	8.07	[46]
**ZIF-8/MPPM**	**0.151**	**1.96**	**11.50**	**This work**

**Table 2 membranes-13-00500-t002:** The flexibility of polymer-supported ZIF-8 membranes compared.

Membrane and Substrate	Performance Retention Curvature	Reference
ZIF-8/PSF	36.3 m^−1^	[21]
ZIF-8/porous polyether sulfone fiber	46 m^−1^	[22]
ZIF-8/PP	92 m^−1^	[23]
**ZIF-8/MPPM**	**348 m^−1^**	**This work**

## Data Availability

The data presented in this study are available in this article.

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
