# Peer review of "Preparation and Lithium-Ion Separation Property of ZIF-8 Membrane with Excellent Flexibility"

_membranes, 2023, doi:10.3390/membranes13050500_

Round 1

Reviewer 1 Report

Formation composite membranes based on ZIF-8/MPPM for selective Li+ separation.

The authors first obtained a zeolite membrane with good flexibility and investigated it in an important practical area - the extraction of lithium from salt water. The authors presented a very interesting and relevant article on the isolation of lithium from saline solutions. It would be more interesting to compare the properties of membranes on a Li/Na pair than Li/Mg. Can the authors supplement the second part of the article (influence of morphology) with data on the pair Li/Na?

In the text of the discussion of the results there is a comparison with the literature data. Also in the introduction, the nearest analogues of the studied membranes are considered in detail. Of course, this part of the work can always be expanded and deepened, but the article looks concise. The main advantage of the studied membranes, the authors note their mechanical properties. Of course, a more visual representation of the comparison with existing literature data would have made it possible to more clearly identify the novelty and significance of this study.

It would be interesting if the authors conducted a comparative analysis of their membranes with standard/commercial samples, including a direct method to evaluate their mechanical properties. It would strengthen the article. It is worth supplementing the conclusions with a description of the novelty of the study and the significance of the results obtained. In my opinion, it is worth adding to Fig. 7 and 8 are given for sodium chloride.

Reviewer 2 Report

This manuscript reported a flexible ZIF-8 membrane was synthesized successfully on microporous polypropylene film as a support. I think the membrane preparation method and the membrane flexibility are interesting for readers. However, further clarification and justification of some statements should be made. So, I recommend the publication of this manuscript after revision. Specific comments are below.

1.        The author mentioned “Thanks to their regular pore structure as well as precisely adjustable pore size, MOFs are ideal film-forming materials for designing high-quality lithium ion separation membranes [12].” on page 1, lines 41-42. I agree that MOFs are one of the applicable membrane materials for lithium-ion separation due to pore structure and adjustable pore size. However, I could not find the statement regarding “MOFs are ideal film-forming materials for designing high-quality lithium ion separation membranes” in reference 12. Please let me know where the statement is written in reference 12.

2.        I think “ion flux” or “ion permeation flux” is the correct technical term for Ji of Equation (1) instead of “ion permeation rate”. Besides, the author used “permeability” with the same meaning as “ion permeation rate” in Figures 6, 7, and 8. The author should use the same term consistently in the manuscript.

3.        According to Figure 2d, the thickness of the ZIF-8 polycrystal layer seems approximately 50 to 80 μm. Does this membrane prepare under the standard condition described in the Experimental section? Which is the same condition in Figures 3a-e?

4.        The author should add the concentration of sodium formate and Zn2+ to Figure 3 or the caption of Figure 3. Does (a), (b), (c), (d), and (e) correspond to 0.05, 0.1, 0.125, 0.15, and 0.2 M in Figure 3 regarding the concentration of sodium formate? The author explained Figure 3a-e on page 5, lines 182-191. However, the author only made a qualitative rather than quantitative discussion of crystal size. So, the author should discuss the crystal growth with values of crystal size shown in Figure 3. I recommend the author add SEM images for the cross-section of ZIF-8/MPPM if you discuss thickness.

5.        Regarding Figure 7a-d, please explain the synthesis condition and the properties of each membrane. I could not understand whether membranes were the same condition or not for comparison with pH and temperature.

6.        According to Figure 7, why does the permeability of Li+ decrease by a tenth of the unary permeability shown in Figure 6a? The Li+/Mg2+ selectivity seems low about 5 to 15. Where does this membrane performance stand in comparison to the literature?

7.        Does curvature mean the reciprocal of the radius of the cylindrical bar used to bend the membrane? If so, the higher curvature, the more flexible. How about the curvature of MPPM? Moreover, are there any polymers with a curvature similar to your ZIF-8 membrane? I think if the ZIF-8 membrane can be compared with other polymers, it would give a better idea of how flexible it is for the readers.

8.        The scale bars are unclear. Please revise Figure 8d-f.

9.        In conclusion on page 10 lines 295-296, it mentioned “The ion selectivity order of ZIF-8/MPPM is as follows: Li+>Na+>Mg2+.”, but isn’t it “The ion permeation order of ZIF-8/MPPM is as follows: Li+>Na+>Mg2+.”?

10.    There are some typos and small mistakes as follows, so please revise them.

a)        Revise from “ofZIF-8” to “of ZIF-8” on page 2, line 110.

b)        Revise from “as2-methylimidazole” to “as 2-methylimidazole” on page 2, line 110.

c)        Please use the same unit for hour throughout the manuscript. For example, please confirm “shaken lasting 12 hrs” on page 3, 101 and “deionized water for 6 h” on page 3, 102.

d)        Revise from “Figure 2cdepicts” to “Figure 2c depicts” on page 4, line 169.

e)        Revise from “ZIF-8crystals” to “ZIF-8 crystals” on page 5, lines 181 and 183.

f)         Add a space between “FTIR.” and “as” on page 6, line 203.

g)        Add a space between the axis name and unit of all x- and y-axises in Figure 5 on page 7.

h)        I suggest adding (Dh) after “hydrated ion diameter sizes” on page 7, line 231.

i)         Delete a dot from the unit “mol m-2. h-1” on page 7, lines 229-230.

j)         Revise from “Na+” to “Na+” on page 7, line 239.

k)        Please revise from d) to (d) in the caption of Figure 7 on page 9, line 271.

l)         Please use the same term for selectivity in this manuscript. For example, the selectivity and separation ratio in the sentence “the selectivity of ZIF-8/MPPM for Li+/Mg2+ was reduced, but the separation ratio still reached 5.31”. on page 9, lines 282-283.

m)     Please revise reference 12 from “341, (6149), 974-+.” to “341, 1230444.” on page 11, line 332.

Round 2

Reviewer 2 Report

The revised manuscript has improved figures and captions, making it easier to understand the content. However, there are still some unclear points. So, I recommend the publication of this manuscript after revision. Please confirm the attached file.
